# Experimental Study on Drilling MDF with Tools Coated with TiAlN and ZrN

**DOI:** 10.3390/ma12030386

**Published:** 2019-01-26

**Authors:** Krzysztof Szwajka, Joanna Zielińska-Szwajka, Tomasz Trzepiecinski

**Affiliations:** 1Faculty of Mechanical Engineering and Technology, Rzeszow University of Technology, Kwiatkowskiego 4, 37-450 Stalowa Wola, Poland; j.zielinska@prz.edu.pl; 2Department of Materials Forming and Processing, Rzeszow University of Technology, Powstancow Warszawy 8, 35-959 Rzeszow, Poland; tomtrz@prz.edu.pl

**Keywords:** drilling MDF, thrust force, cutting temperature, surface roughness, cutting tool coating

## Abstract

There is increasing use of wood-based composites in industry not only because of the shortage of solid wood, but above all for their better properties such as: strength, aesthetic appearance, etc., compared to wood. Medium density fiberboard (MDF) is a wood-based composite that is widely used in the furniture industry. The goal of the research conducted was to determine the effect of the type of coating on the drill cutting blades on the value of thrust force (F_t_), cutting torque (M_c_), cutting tool temperature (T) and surface roughness of the hole in drilling MDF panels. In the tests, three types of carbide drills (HW) were used: not coated, TiAlN coated and ZrN coated. The measurement of both the thrust force and the cutting torque was carried out using an industrial piezoelectric sensor. The temperature of the cutting tool in the drilling process was measured using an industrial temperature measurement system using a K-type thermocouple. It was found that the value of the maximum temperature of the tool in the drilling process depends not only on the cutting speed and feed rate, but also on the type of coating of the cutting tool. The value of both the cutting torque and the thrust force is significantly influenced by the value of the feed rate and the type of drill coating. The effect of varying plate density on the surface roughness of the hole and the variation of the value of the thrust force is also discussed. The results of the investigations were statistically analyzed using a multi-factorial analysis of variance (ANOVA).

## 1. Introduction

Medium density fiberboard (MDF) is a wood-based product widely used in the furniture industry [1,2,3,4,5]. MDFs are composed of wood fibers, bonded with formaldehyde glue under the influence of heat and pressure. The use of medium density fiberboards in industry is associated with their machining during furniture production. One of the most commonly used operations in the production of MDF furniture is drilling. The MDFs machinability is determined by the quality of the surface [5,6], which largely depends on the degree of tool wear and the mechanism of chip formation [7]. Various studies have been carried out to improve understanding of MDF cutting characteristics [4,8,9,10]. Most of the studies are mainly focused on measuring the cutting forces and the friction phenomenon based on the theory used in the cutting process of metals [11,12]. Cutting forces, temperature, and surface roughness that reflect susceptibility to material processing are three important issues in the machining of wood-based materials. Cutting forces have a direct effect on energy consumption, tool wear, heat generation and the quality of the surface machined [13,14,15].

In order to maintain the relatively long service life of cutting tools used for MDF machining, it is necessary to use tools with a high wear resistance. Blades made from HM (ISO Code K20) cemented carbide or diamond are the most commonly used in the machining of wood-based materials. In the literature, we can find the results of investigations aimed at increasing the durability of the cutting tool blades in the machining of wood-based composites. Numerous experiments are focused on increasing the durability of the cutting tool blades by applying various coatings to the blade surfaces in order to reduce their wear [16,17,18,19,20,21,22]. Recently, this field of research has been developing dynamically. However, the use of cutting tools with new coatings requires a number of tests, within which the quality of the surface machined, the cutting temperature and the cutting resistance are assessed. The drilling process in metals has been widely studied, and the results of these tests are well described in the literature, while the process of drilling MDF has not received much attention [5,15,23,24].

The cutting process of wood-based materials is strictly dependent on their physical and mechanical properties. MDF has a more homogeneous structure than solid wood. While solid wood exhibits anisotropic properties, MDF consists of several isotropic layers [25] where the highest density is observed at the edges of the panel, and the lowest density is in the middle of the panel.

Gordon and Hilery [10] presented a brief overview of work on forecasting the cutting forces in MDF machining. These works described the general mechanics of MDF machining based on assumptions occurring in metalworking. Djouadi et al. [17] investigated the use of polycrystalline diamond (PCD) cutting blades in MDF machining. They concluded that the main benefit of using PCD is the extended durability of the cutting tool resulting from its greater hardness and better tribological properties compared to traditional tool materials. Davim [2] studied the effect of various cutting parameters on surface roughness in the MDF milling process using uncoated cutting tools. Similar studies were carried out by Sedlecký [23].

Szwajka and others [15] conducted investigations of the drilling of melamine faced chipboard using a carbide drill. They developed an analytical model for predicting the effect of drilling parameters on thrust force. It was found that an increase in the value of the feed rate increases delamination of the chipboard.

The quality of the MDF surface is an important factor influencing the final appearance of the product or subsequent technological processes, such as gluing (adhesion and cohesion), coating, varnishing, etc. [23,26]. The surface roughness of the surface machined depends on various factors and conditions [2,24,27], which can be classified as follows: type of machining (cutting, milling, drilling, etc.) [28,29,30], machining parameters (cutting speed, feed rate) [31,32,33,34], type of cutting tool (geometry, applied coatings on the cutting edge), as well as the properties of the material being processed [26].

Even if the machining parameters are the same, each machining method leaves characteristic irregularities on the surface; for example, saw cut surfaces differ from milled surfaces [8,35]. The requirements for surface roughness are determined in accordance with the functional application of the future product [36,37]. The surface roughness is determined by specifying the numerical value of one or several surface roughness parameters and the value of the sampling length [23].

Lin et al. [36] analyzed the machinability of MDFs. They used a digital camera to record chip formation occurring in front of the cutting tool edge and a scanning electron microscope (SEM) for additional analysis of the surface machined. Davim et al. [38] studied the the cutting parameters (cutting velocity and feed rate) under specific cutting pressure, thrust force, damage and surface roughness in glass fiber reinforced plastics. It was shown that the differences in MDF panel density are closely related to machinability characteristics.

The heat generated in the drilling process has a direct effect on the surface roughness of the surface machined, hole quality and chip morphology [1,39]. The temperature in the region of contact of the blade with the material processed (wood or wood-based material) depends both on the energy released in this region and the efficiency of heat removal in this region. In addition to radiation, conductivity is the main mechanism responsible for heat removal. Considering the fact that the thermal conductivity of wood and wood-based materials is very low, and the use of cooling liquids is excluded, the main element responsible for heat removal is the tool. This leads to an undesirable increase in tool blade temperature. During the drilling process, the most important factor affecting the cutting output is the temperature created between the cutting tool and the workpiece. In this paper, the effect that the type of coating on the drill cutting blades has on the value of thrust force, cutting torque, cutting tool temperature, and surface roughness of the hole in drilling MDF panels is investigated. Three types of carbide drills (HW) were used in the tests: not coated, TiAlN coated and ZrN coated.

## 2. Experimental Procedure

### 2.1. Material

A typical industrial MDF panel with a thickness of 18 mm was used as the workpiece. The mechanical and physical properties of the material being processed are listed in Table 1. An MDF panel is characterized by a clear differentiation of material density in the cross-section, which results from its multi-layered structure. To more accurately characterise the workpiece, a laboratory measurement of the density profile through the panel thickness was carried out using a Phoenix v-tomome X-ray tomograph (GE Sensing & Inspection Technologies, Wunstorf, Germany).

X-ray tomographs permits one to obtain tomographic images of the object examined, and then present its spatial (3D) image from many flat (2D) images taken in various positions. The computer tomographic (CT) images contain information about the location and density of the absorbing features in the object. Any difference in material density inside the object can be measured and visualised. Figure 1 shows a picture of the cross-section of an MDF panel, in which a distinct variation in plate density can be observed. The highest density occurs in the outer layers of the panel (to a depth of about 2.3 mm). However, a lower density appears in the inner layer (around 13.4 mm in length) (Figure 1).

Furthermore, the hardness distribution of the material processed was measured using a Shore hardness tester (Hildebrand, Oberboihingen, Germany) using the Shore D scale (Figure 2b). As can be seen, the hardness distribution (Figure 2a) is closely related to the density profile. The highest hardness value occurs in the outer layers with a thickness of approximately 2.3 mm and is equal to 62 °Sh (D scale). As we move away from the outer layer, the hardness decreases, reaching a value of 43 °Sh (D scale) at a depth in the range between 7 and 11 mm.

The spectral analysis of the elements constituting the material (Figure 3) was carried out using a scanning electron microscope (TESCAN, MIRA3, Brno, Czech Republic).

### 2.2. Cutting Tools

In the drilling of MDFs, HW sintered carbide cutting tools were used with different types of cutting blade coating. Two types of coating for cutting blades, i.e., TiAlN and ZrN were used. The selection of coatings was dictated by the fact that they belong to those most commonly used in the machining of composite materials [40]. In the case of cutting wood-based materials, there is a general lack of commercially available cutting tools (especially drills) with additional protective coatings. In the research, it was necessary to measure the temperature of the cutting blade during the drilling process. To measure the temperature between the cutting edge and the workpiece using thermocouples, it was necessary for the tool to have coolant channels. However, no cooling liquids are used in the treatment of MDFs. Therefore, it was necessary to fabricate drills. The geometry of the drills was based on the geometry of the Leitz^®^ HW/D10/NL35/S10x24/GL70 drill (Leitz GmbH & Co. KG, Oberkochen, Baden-Württemberg, Germany), which is widely used in the drilling of through-holes in MDFs.

A coordinate measuring machine is used to measure the geometrical dimensions of the reference tool in order to make drills with coolant channels identical to the standard ones but made of cemented carbide monolith, which will be covered with two different coatings. The Zoller^®^ Genius 3 coordinate measuring machine (Zoller, Pleidelsheim, Germany) (Figure 4) was used to measure and control the geometry of the cutting tools. This machine is used by tool manufacturers to enable one to take measurements in a fully automatic way. Fully automatic precise measurements are assured by five numerically controlled axes (X, Y, Z, C, B).

The Genius 3 device is equipped with an optical system of two cameras: a main camera for measurement in transmitted and reflected light with a magnification of 50× and a tilt camera for measuring only in reflected light with a lens capable of focusing in 3D mode with a magnification of 200×. To accurately capture the details of the cutting blade, the device is equipped with eight-segment automatically tuned LED lighting.

The computer-aided design of the tool was developed in the Numroto^®^ Plus program (NUM AG, Teufen, Switzerland). Numroto Plus software is commonly used to design rotary cutting tools and to generate a 3D model (Figure 5c) after prior definition of all the geometrical values of the cutting tool, the selection of the type of grinding wheels and workpiece size, and the setting of machining parameters on a numerically controlled grinder. The grinding wheels used in the fabrication of drills are Toolgal^®^ diamond grinding wheels (TOOLGAL Industrial Diamonds Ltd., Degania, Israel). Two 1A1 grinding wheels and two types 11V9 and 12V9 were used for the fabrication of drills.

The finished project made in the Numroto^®^ Plus program was exported to the 5-axes grinder from a Saacke^®^ model UW IF (SAACKE GmbH & Co. KG, Pforzheim, Germany), shown in Figure 5a,b.

An effective way to increase the durability of cutting tools is to apply a coating on their blades. The most commonly used method is Physical Vapour Deposition (PVD) in which the solid metal is vaporised in a high vacuum environment and deposited in the form of a thin layer on the tool surface. The deposited layer, usually with a thickness of 3–5 μm, has a very high hardness, usually in the range of 2000–3000 HV, which significantly increases the resistance of the tool blades to abrasive wear. The machined tools (Figure 6b) were PVD coated with zirconium nitride (ZrN) and aluminum titanium nitride (TiAlN) using an EIFELER VACOTEC PVD Alpha 400 vacuum reactor (Vacotec GmbH, Düsseldorf, Germany) (Figure 6a).

The coatings were made on a substrate previously sprayed with high energy ions, free of oxides and enriched with elements forming a strong adhesion-diffusion bond. The selected properties of the coatings used are listed in Table 2.

Three types of drills with coolant channels were fabricated: an HW carbide drill with a ZrN coating, an HW carbide drill without a coating and an HW carbide drill with a TiAlN coating (Figure 6b). The spectral analysis of the elements included in the coating applied (Figure 7) was carried out using a TESCAN^®^ scanning electron microscope (TESCAN, MIRA3, Brno, Czech Republic).

### 2.3. Equipment and Machining Conditions

The drilling process was carried out in two stages: on a CNC vertical milling machine and on an EMCO^®^ CNC lathe (EMCO GmbH, Hallein, Austria). A schematic diagram of the configuration of the measurement path and the measurement data archiving system is presented in Figure 8. In the first stage of testing, the CNC milling machine recorded the values of thrust force (F_t_) and cutting torque (M_c_) during MDF machining.

In the first stage, three holes (with the same set of cutting parameters) were drilled in an MDF panel with dimensions 130 mm × 30 mm × 18 mm on a CNC milling machine. The value of the thrust force and the cutting torque was measured using the Kistler^®^ 9345B2 piezoelectric industrial sensor (Kistler, Winterthur, Switzerland). The signals from the sensor were recorded on a personal computer (PC) disk via the National Instruments^®^ 6034E (National Instruments Corporation, Austin, TX, USA) 16-bit analogue-to-digital card with a sampling rate of 50 Hz. The surface topography of each completed hole was measured in two locations (every 180°) using the Hommel-Etamic T8000RC CNC profilometer (Jenoptik, Jena, Germany) (Figure 9). In order to measure the surface roughness, the test sample was cut into two parts along the hole axis. One measurement was made for each part separately.

In the second stage of the investigations, the holes were drilled (with the same set of cutting parameters) in MDF workpieces with a diameter of 30 mm and a thickness of 18 mm on a CNC lathe. The temperature value was measured during machining between the cutting edge and the workpiece using the National Instruments^®^ 9212 industrial system (National Instruments Corporation, Austin, TX, USA). Two K-type thermocouple wires with a diameter of 0.2 mm were used for temperature measurement. The thermocouple wires were mounted in the cooling liquid drill channels (Figure 10). Signals from the measurement system were recorded in digital form on a personal computer (PC) disk. The sampling rate of signals during experiments was 5 Hz, and a 24-bit measurement card was used.

The cutting parameters used during the drilling experiments are listed in Table 3. Three replications were made for each of the sets of cutting parameters. This research methodology was applied to drilling with the following types of drills: not coated, ZrN coated and TiAlN coated.

In order to avoid the accidental influence of the radius of the cutting edge of drills (ε_r_) on the machinability indicators (thrust force, cutting torque, temperature and surface roughness), the drills with the same value of the cutting edge radius were selected. The variation of the the cutting edge radius ε_r_ of drills used varied between 5.91 and 5.95 μm. The Zoller Genius 3 measuring machine (Zoller, Pleidelsheim, Germany) (Figure 4) was used to measure the ε_r_ radius. Figure 11 shows the details of measurement of radius of the drill’s cutting edge.

## 3. Results

### 3.1. Feed Force and Cutting Torque

Figure 12 shows the variation in thrust force, cutting torque and cutting edge temperature as a function of cutting length when drilling the MDF at a cutting speed of 35 m/min and a feed rate of 167 mm/min using the TiAlN coated drill.

It was observed that five major phases of the variation of the thrust force (F_t_) value can be identified during drilling (Regions I–V in Figure 12). In the period referred to as Phase I, the chisel edge of the drill penetrates into the material producing a rapid increase in the value of the thrust force. During this phase, instead of cutting, the chisel edge of the drill is pressed into the material. Then, the drill starts to cut the outer layer of the material with the highest density and hardness. When the chisel edge of the drill leaves the area of material with the highest density (depth approx. 2.2 mm), the force value decreases due to the fact that the drill sinks into the middle layer of the panel with lower density and hardness. During Phase II, the drill penetrates to a depth of h = 8.1 mm, which is equal to the height of the cutting blades of the drill (Figure 5c). During this time, there is an increase in the cross-sectional area of the cutting layer, which results in a proportional increase in the thrust force. Phase III corresponds to cutting with a constant cross-section of the cutting layer in the middle layer of the panel. Stabilisation of the thrust force value is then observed. At the beginning of Phase IV, the value of the thrust force starts to increase rapidly due to the fact that the drill begins to penetrate the outer layer of the material (higher density and hardness). The thrust force reaches its maximum value. When the chisel edge of a drill leaves the workpiece (cutting path approx. 18 mm), the thrust force starts to decrease (Phase V). The value of the thrust force drops to zero at the end of Phase V when the cutting edges of the drill have left the workpiece.

In the case of cutting torque (M_c_), it is clearly visible in Phase 1 that there is no machining, and only the chisel edge of a drill is pressed into the workpiece. In Phase I, the cutting torque value is close to zero. Next, the blades sink into the workpiece (Phase II) which results in a proportional increase in the value of cutting torque. The period of this increase continues until the drill reaches the maximum cross-sectional area of the cutting layer, i.e., the drill is penetrated to a depth of h = 8.1 mm. In Phase III, the cutting of the cutting layer with a constant cross section in the middle layer of the panel proceeds. A stabilisation of the cutting torque value is observed. At the beginning of Phase IV, the value of the cutting torque starts to increase due to the fact that the drill begins to enter into the outer layer of material (higher density and hardness). Furthermore, in this phase, the largest cross-section of the material is cut, and the cutting torque reaches its maximum value. When the drill begins to come out of the workpiece (cutting path approx. 18 mm), the cutting torque starts to decrease (Phase V). In the middle part of Phase V, some stabilisation of the cutting torque is observed. As a result, the drill that comes out of the material cuts only the outer layer of the material characterized by high density and hardness. The cutting torque value drops to zero at the end of Phase V when the cutting edges of the drill have left the workpiece. For the analysis of the acquired signals of F_t_ and M_c_, an individually designed computer program was prepared in the LabVIEW programming language enabling, at selected time intervals, the mean values of the recorded thrust force and cutting torque signals to be determined. The program was based on the automatic determination of values of F_t_ and M_c_ parameters in a specific time range of the signal. The methodology for determining the average values of recorded signals has been described in detail in [5]. Figure 13a–c shows the effect of feed per revolution (f) on the thrust force (F_t_) for the three cutting speeds and the three types of drill coating. The thrust force values presented in the graphs were obtained as the average result of three repetitions.

Multi-factorial analysis of variance (ANOVA) carried out in the STATISTICA program allowed the verification of the significance of the influence of several independent variables on the dependent variable. Furthermore, multivariate analysis makes it possible to take the synergistic effect of the product of many variables into account in the statistical model. Taking into account the adopted level of significance of *p* = 0.05, the statistical significance of particular groups of variables and individual variables is determined. The results of the analysis (Table 4) allow one to reject, at a significance level *p* = 0.000, the hypothesis concerning the lack of effect of the factors “coating”, feed per revolution (f) and cutting speed (v_c_) on the value of the thrust force (F_t_). There was no statistically significant effect of interactions between the factors analyzed.

Figure 13d presents a comparison between the results for the values of the thrust force obtained during the experiment with the values obtained based on the analytical models (1, 2, 3). The correlation coefficients obtained were equal as follows: R^2^ = 0.990 for the TiAlN coated drill, R^2^ = 0.983 for the ZrN coated drill and R^2^ = 0.993 for the uncoated drill.

The smallest value of thrust force (F_t_) was obtained in the drilling process using a tool without a coating. However, the highest values of thrust force were obtained for a drill with a TiAlN coating. The increase in the value of the thrust force in comparison to machining without a coating was on average about 39%. In the case of a ZrN coated drill, the value of the thrust force obtained was reduced in comparison to the TiAlN coated drill, but it was still higher on average by about 17% in relation to the value obtained with the use of a drill without a coating. For example: for the cutting speed v_c_ = 105 m/min and feed per revolution value f = 0.2 mm/rev the value of the thrust force for the TiAlN coated drill was 33 N, for the tool with a ZrN coating was 27.8 N and for the tool without a coating it was 23.7 N. This can be explained by the differing values of the coefficient of friction between the tool and the workpiece material resulting from the type of coating used. For all the coatings used, the value of the thrust force increases with an increase in the value of the feed per revolution. The value of the thrust force can be described by the Equations (1)–(3).
F_t_ (N) = 25.139 + 52.333 × f − 0.021 × v_c_, (ZrN)(1)
F_t_ (N) = 20.005 + 50.333 × f − 0.018 × v_c_, (TiAlN)(2)
F_t_ (N) = 13.011 + 66.667 × f − 0.026 × v_c_, (uncoated)(3)

Figure 14a–c shows the relations of cutting torque (M_c_) versus the feed per revolution (f) for the three cutting speeds (v_c_) and three types of drill coatings. The influence of these parameters on the cutting torque (M_c_) is noticeable to a lesser extent.

Figure 14d shows a comparison of the results of the cutting torque values obtained during the experiment with the values obtained on the basis of the analytical model.

The results of the statistical analysis (Table 5) allow one to reject, at the level of significance *p* = 0.000, the hypothesis that the coating type, feed per revolution (f) and cutting speed (v_c_) do not affect the value of the cutting torque (M_c_). In the case of interactions between the factors analyzed, no statistically significant effect of these factors on the value of (M_c_) was observed.

The lowest value of torque was obtained using the drill without a coating in the drilling process (Figure 14). The highest values of cutting torque (M_c_) recorded in the experiments were obtained using a drill with a TiAlN coating. The increase in the value of cutting torque when machining using coated drills in comparison to that using an uncoated drill was approximately 35%. In the case of a ZrN coated drill, the value of the cutting torque (M_c_) obtained was less than that using a TiAlN coated drill, but it was still greater by approx. 15% in relation to the value of (M_c_) obtained with a drill without a coating.

For example: for the cutting speed v_c_ = 35 m/min and feed per revolution value f = 0.2 mm/rev, the cutting torque (M_c_) for the TiAlN coated drill was 0.35 Nm, for the ZrN coated drill it was 0.3 Nm and for the drill without coating was 0.26 Nm. In a similar manner to the effect of drilling parameters on the value of the thrust force, this fact can be explained by the differing values of the coefficient of friction between the tool and the workpiece resulting from the type of drill coating. For all the coatings used, the value of the thrust force (F_t_) increases with an increase of the feed per revolution (f) value. The value of the thrust force (F_t_) can therefore be described by the Equations (4)–(6).
M_c_ (Nm) = 0.218 + 0.607 × f − 0.001 × v_c_, (ZrN)(4)
M_c_ (Nm) = 0.167 + 0.557 × f − 0.001 × v_c_, (TiAlN)(5)
M_c_ (Nm) = 0.093 + 0.747 × f − 0.001 × v_c_, (uncoated)(6)

The correlation coefficients obtained between the experimental and predicted values of cutting torque M_c_ (Figure 14d) were equal as follows: R^2^ = 0.981 for a TiAlN coated drill, R^2^ = 0.997 for a ZrN coated drill and R^2^ = 0.993 for an uncoated drill. For all the coatings of tools used, the value of the cutting torque (M_c_) increases as the feed per revolution (f) increases.

### 3.2. Analysis of Surface Topography

Surface topography is one of the main features taken into account to evaluate the surface quality in machining processes. The value of the roughness average Ra parameter was measured in the longitudinal direction of the holes machined in MDF panels. The choice of this indicator to assess surface roughness was dictated by its very frequent use in production plants [23]. The value of the roughness average (Ra) was determined on the basis of the surface topography map in selected measurement sections. Figure 15a–c shows an example of the topography of the hole surface obtained in the process of drilling with a TiAlN coated drill at a cutting speed of 105 m/min and with three feed per revolution values.

Three areas with a different character of surface roughness can clearly be seen on the surface topographies presented. The first and the second area occur in the outer layers of the MDF panel and the third area in the middle layer of the board. This diversity can be explained by the abovementioned multi-layered structure of the MDFs. The resulting surface topography accurately reflects changes in both hardness and panel density.

Measurement of the roughness average (Ra) parameter was carried out in accordance with the recommendations of ISO-4288:2011. The test conditions of the surface roughness measurement were adopted in accordance with Table 6. The mean groove spacing (RSm) value (Figure 16) in the measurements was in the range between 0.13 and 0.4.

The value of the Ra roughness average parameter was determined separately for the outer layer and middle layer of the MDF. Therefore, the additional effect of both the density and hardness of the workpiece on the Ra parameter can be analyzed.

Figure 17 and Figure 18 show the effect of the feed per revolution value for the three cutting speeds and all types of tool coatings. The values of roughness average (Ra) in Figure 17 and Figure 18 are the average of six measurements. As mentioned, two measurements were made for each hole, and the drilling of each hole was repeated three times. The results presented in Figure 17 refer to the measurement of the surface roughness in the middle layer of the workpiece.

Figure 18 presents the results regarding the measurement of surface roughness in the outer layer of the workpiece. In both cases, as the cutting speed (v_c_) increases and simultaneously the feed per revolution (f) decreases, the value of the roughness average (Ra) decreases. For the inner layer of panel at a cutting speed of 35 m/min and a feed per revolution of 0.2 mm/rev, the roughness average parameter was Ra = 10.3 μm for a TiAlN coated drill, for a cutting speed of 105 m/min and a feed per revolution of 0.1 mm/rev, the roughness average parameter was Ra = 6.6 μm. However, for the outer layer of the panel at a cutting speed of 35 m/min and a feed per revolution of 0.2 mm/rev, the roughness average parameter was Ra = 5.1 μm, while for a cutting speed of 105 m/min and a feed per revolution of 0.1 mm/rev the roughness average parameter was Ra = 3.3 μm for a drill with TiAlN coating. This can be explained by the fact that the accumulation of chips in the chip spaces decreased with increasing cutting speed (v_c_). In addition, the very pronounced impact of the type of tool coating on the surface roughness value was noted. It has been found that machining with a ZrN coated tool allows one to achieve the lowest surface roughness value compared to a TiAlN coated tool and an uncoated tool. This may be caused by a different value of friction coefficient as well as of thermal conductivity coefficient depending on the type of coating (Table 2). A higher value of friction coefficient and a lower value of coefficient of thermal conductivity causes an increase in the temperature value in the tool-workpiece contact area. The increase in heat generated in the area of contact between the cutting tool and the workpiece, in the case of MDF, significantly improves the connection of wood fibers and formaldehyde adhesive. It causes the compaction of the bonds between fibers.

The results of the analysis (Table 7) allow one to reject, at a significance level *p* = 0.000, the hypothesis that the coating type and feed per revolution (f) do not affect the Ra parameter value. There was no statistically significant impact of the cutting speed (v_c_) on the Ra parameter value. Similarly, in the case of interactions between the factors analyzed, no statistically significant impact was observed.

The experiments conducted showed that the feed per revolution (f) value, cutting speed (v_c_) and the type of tool coating have a significant influence on the roughness average (Ra) parameter. However, the Ra parameter values measured in the outer layers are significantly lower than those measured in the inner layer.

### 3.3. Temperature of Cutting Tools

The highest values of tool temperature were observed in the case of the ZrN coated drill, and the lowest in the case of the uncoated drill (Figure 19a–c). The increase in the temperature value for a ZrN coated drill compared to a drill without a coating was about 20%. In the case of the TiAlN coated drill, the value of cutting torque obtained (M_c_) is lower compared to the ZrN coated drill, but it was still higher by approximately 13% in relation to the temperature value obtained with the use of an uncoated drill.

In the case of the ZrN drill, the maximum temperature was 56.3 °C. The smallest value of temperature obtained in the drilling process relating to an uncoated drill was 38.5 °C. This fact can be explained by differentiation in both the coefficient of friction between the tool and the workpiece as well as in the value of the thermal conductivity coefficient resulting from the type of tool coating. The ZrN coating has a much lower heat transfer coefficient compared to the TiAlN coating (Table 2). This causes the ZrN coating to be a barrier to removing heat from the cutting zone. In addition, the MDF panel is characterized by a relatively low value of thermal conductivity.

The results of the statistical analysis (Table 8) allow us to reject, at a significance level *p* = 0.000, the hypothesis that the coating type and feed per revolution (f) do not affect the temperature (T). At a significance level *p* = 0.006 there is also a lack of influence of cutting speed (v_c_) on drill temperature. There were no statistically significant interactions between the product factors analyzed. Correlation coefficients of the relation between the experimental and predicted values of tool temperature were equal approximately as follows: R^2^ = 0.918 for a TiAlN coated drill, R^2^ = 0.933 for a ZrN coated drill and R^2^ = 0.941 for an uncoated drill (Figure 19d).

The temperature value of the cutting edge depending on the coating applied is expressed in the form of the Equations (7)–(9):T (°C) = 53.791 − 62.601 × f − 0.065 × v_c_, (ZrN)(7)
T (°C) = 55.978 − 65.241 × f − 0.062 × v_c_, (TiAlN)(8)
T (°C) = 43.991 − 41.801 × f − 0.078 × v_c_, (uncoated)(9)

## 4. Conclusions

This paper presents the results of experimental investigations into the effect of coatings applied to drill blades and cutting parameters on selected indices with regard to MDF machinability in the drilling process. Based on the results obtained, it can be concluded that:In the analysis of the values of thrust force (F_t_), cutting torque (M_c_), and surface roughness parameter (Ra) the layered structure of the MDF panel, which consists of layers of different density and hardness, should be taken into account.A significant effect of the type of drill coating on the value of all the MDF machinability indices analyzed was observed.There is a dominant influence of both the feed per revolution (f) and the type of tool coating on thrust force (F_t_), cutting torque (M_c_) and cutting tool temperature in the MDF drilling process. The highest values of cutting torque (M_c_) and thrust force (F_t_) recorded in the experiments were obtained using a drill with a TiAlN coating. In contrast, the lowest values of M_c_ and F_t_ were obtained using an uncoated drill. This fact can be explained by the differing values of the coefficient of friction between the tool and the workpiece resulting from the type of drill coating. The coated drill surfaces are characterized by the relatively higher friction coefficients than the surface of uncoated ceented carbide tool. Therefore, the coating increases the friction force between the tool and the workpiece, according to the Marchant’s Circle diagram. Increasing in the friction force causes an increase of the cutting resistance.The test results obtained show that the temperature in the drilling process increases with increase in cutting speed (v_c_) but decreases with an increase of the feed per revolution (f) value. Temperature changes, depending on the coating type, vary on average by approximately 20%. The highest values of tool temperature were observed in the case of the ZrN coated drill, and the lowest in the case of the uncoated drill. This fact can be explained as follows: the ZrN coating has a much lower heat transfer coefficient compared to the TiAlN coating and uncoated tool. This causes the ZrN coating to be a barrier to removing heat from the cutting zone.The feed per revolution (f) and the type of drill coating had a significant influence on the value of the roughness average (Ra) parameter. It has been observed that in the outer layers of the panel, the (Ra) parameter value has a lower value compared to that measured in the middle layer. The lowest value of roughness average parameter (Ra) was observed in the case of the ZrN coated drill, and the highest in the case of the uncoated drill. This fact can be explained by a lower value of friction coefficient and a lower value of thermal conductivity coefficient of ZrN coating compared to the uncoated tool. The lower value both of thermal conductivity coefficient and friction coefficient causes an increase in heat generated in the area of contact between the cutting tool and the workpiece. In the case of MDF, this significantly improves the connection of wood fibers and formaldehyde adhesive. It causes the compaction of the bonds between fibers.

In conclusion, the feed per revolution (f) and the type of tool coating are the dominant parameters that significantly affect the drilling process of the MDF board.

## Figures and Tables

**Figure 1 materials-12-00386-f001:**
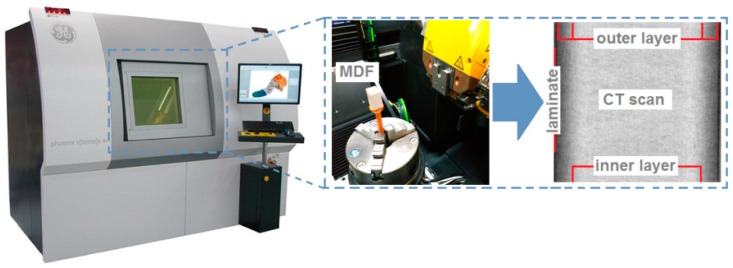
MDF density measurement on a computer tomography Phoenix v|tome|x m.

**Figure 2 materials-12-00386-f002:**
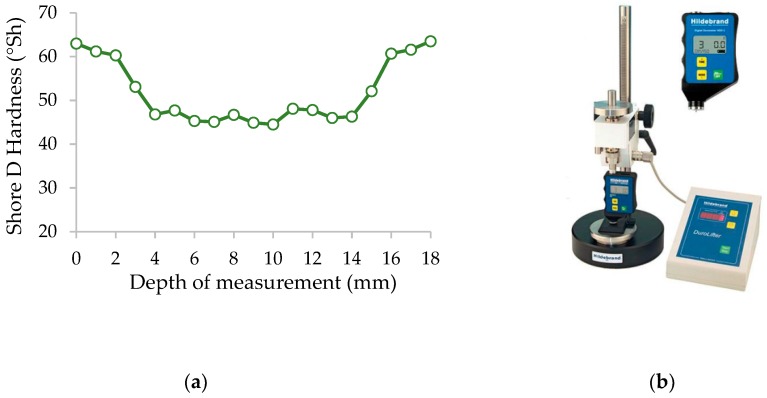
Hardness measurement: (**a**) Hardness profile of MDF used in tests; (**b**) shore hardness tester.

**Figure 3 materials-12-00386-f003:**
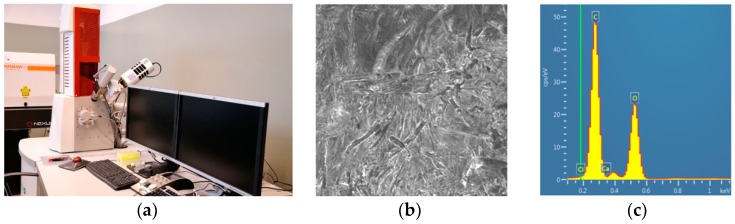
Scanning electron microscope and spectral analysis of the MDF: (**a**) Measuring stand; (**b**) microphotograph of an MDF panel surface; (**c**) spectra of the outer surface of an MDF panel.

**Figure 4 materials-12-00386-f004:**
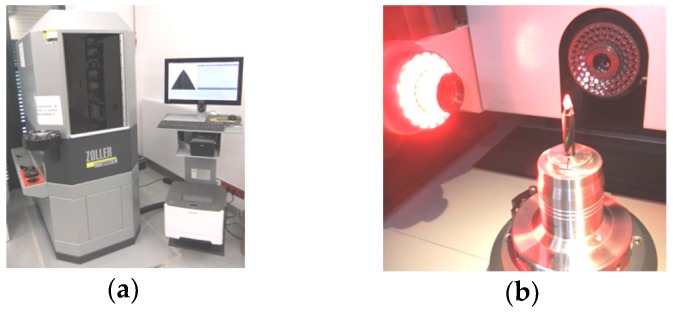
Measuring the cutting tool: (**a**) The Zoller Genius 3 coordinate measuring machine; (**b**) the mounting arrangements and optical system in the Zoller^®^ Genius 3.

**Figure 5 materials-12-00386-f005:**
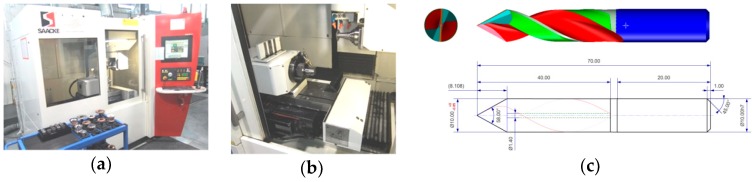
Grinder with drill design: (**a**) SAACKE CNC-Grinding Center Model UW I F; (**b**) work-space of the grinder; (**c**) drill design in the Numroto^®^ Plus program.

**Figure 6 materials-12-00386-f006:**
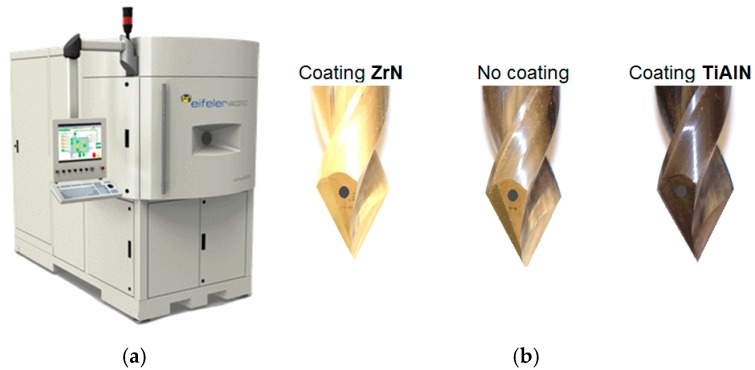
Stand for applying protective coatings: (**a**) Vacuum reactor EIFELER VACOTEC PVD Alpha 400; (**b**) drills with the coatings used in the research.

**Figure 7 materials-12-00386-f007:**
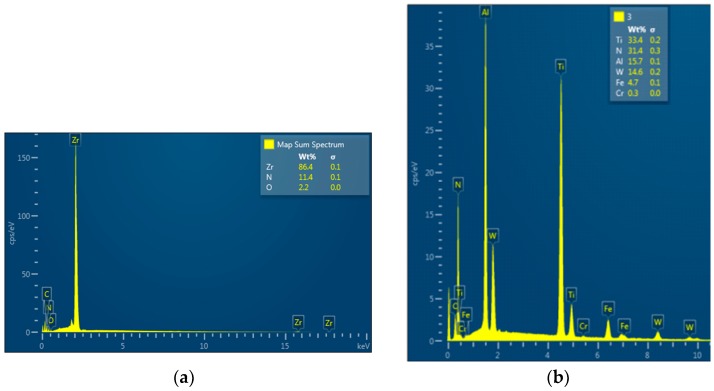
Results of spectral analysis of the drills: (**a**) ZrN coated drill; (**b**) TiAlN coated drill.

**Figure 8 materials-12-00386-f008:**
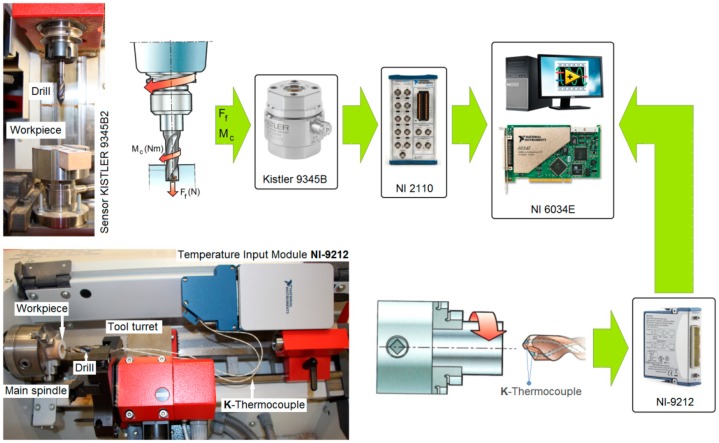
Experimental set-up and schematic diagram of the data acquisition system.

**Figure 9 materials-12-00386-f009:**
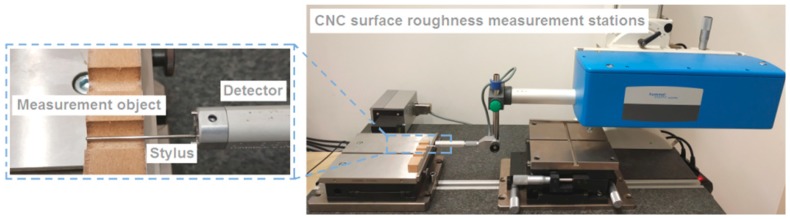
Measurement of surface topography.

**Figure 10 materials-12-00386-f010:**
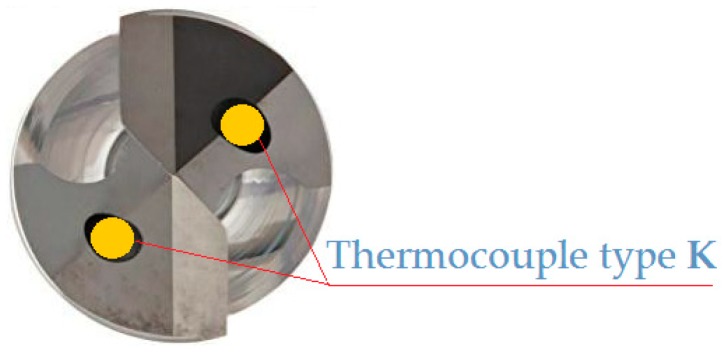
Drill with inserted thermocouples.

**Figure 11 materials-12-00386-f011:**
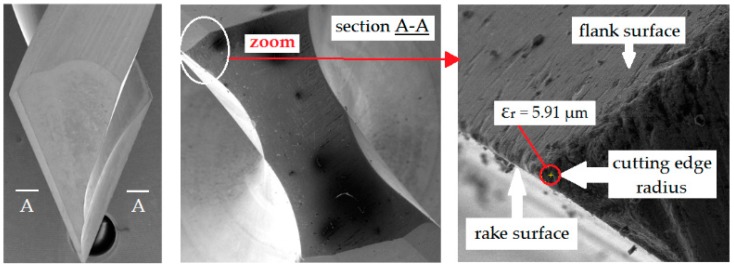
Measurement of radius of the drill’s cutting edge.

**Figure 12 materials-12-00386-f012:**
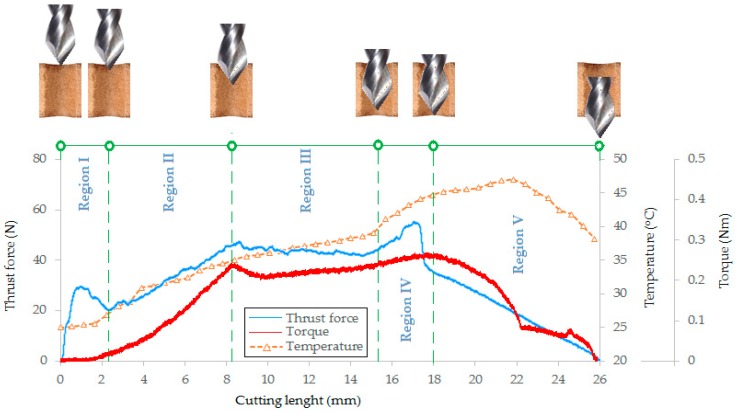
Variation of thrust force, cutting torque and temperature in the drilling process.

**Figure 13 materials-12-00386-f013:**
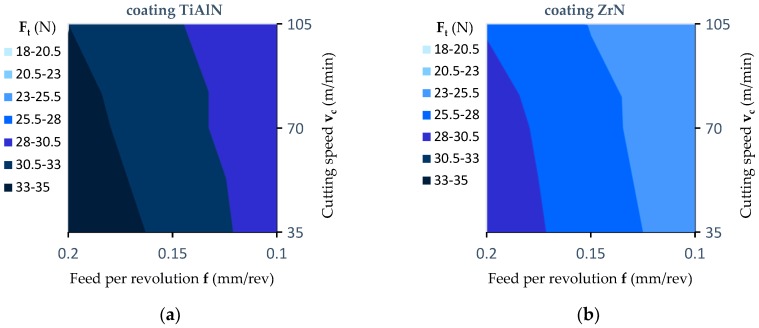
Influence of the feed value on the value of the thrust force (contour diagrams): (**a**) For a TiAlN coated drill; (**b**) for a ZrN coated drill; (**c**) for an uncoated drill; (**d**) correlation between the experimental and predicted values of thrust force determined for the coated and uncoated tools.

**Figure 14 materials-12-00386-f014:**
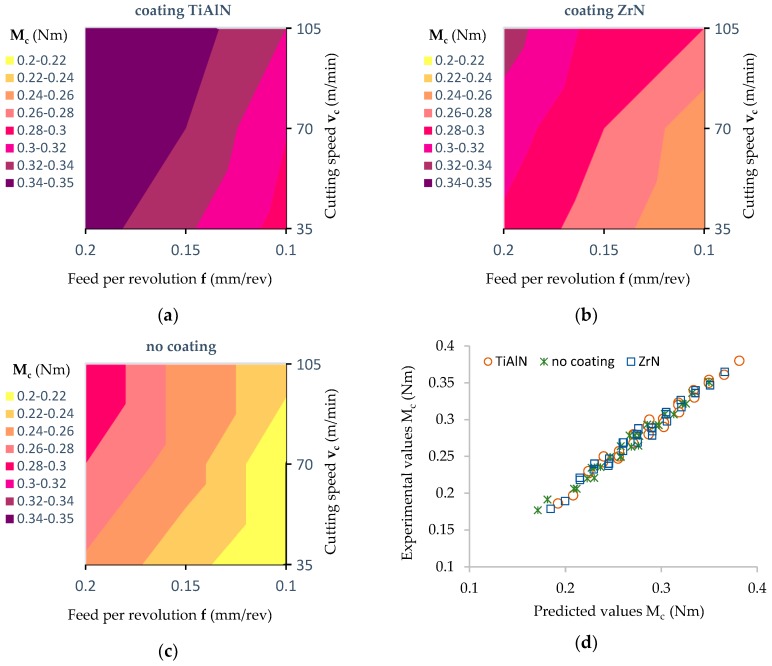
Influence of the feed value on the value of the cutting torque (contour diagrams): (**a**) For a TiAlN coated drill; (**b**) for a ZrN coated drill; (**c**) for an uncoated drill; (**d**) correlation between experimental and predicted values of cutting torque determined for the coated and uncoated tools.

**Figure 15 materials-12-00386-f015:**
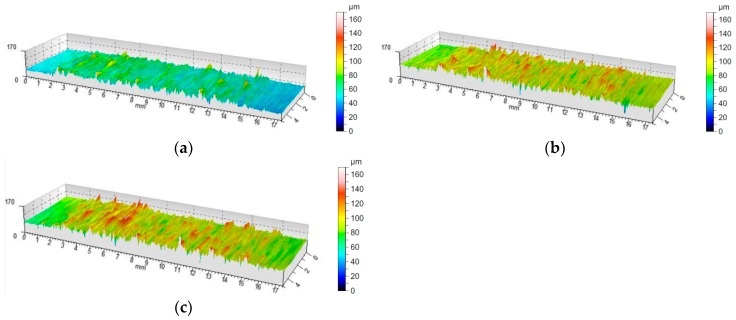
The topography of the surface drilled with a TiAlN coated tool: (**a**) Cutting speed 105 m/min and feed per revolution 0.1 mm/rev; (**b**) cutting speed 105 m/min and feed per revolution 0.15 mm/rev; (**c**) cutting speed 105 m/min and feed per revolution 0.2 mm/rev.

**Figure 16 materials-12-00386-f016:**
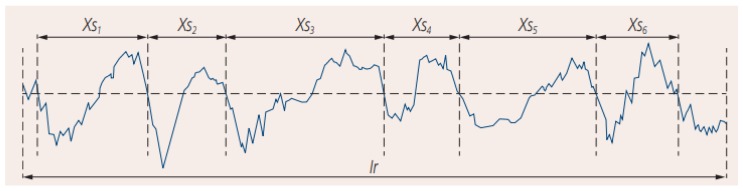
The mean groove spacing RSm as the mean value of the spacing *Xsi* of the profile elements.

**Figure 17 materials-12-00386-f017:**
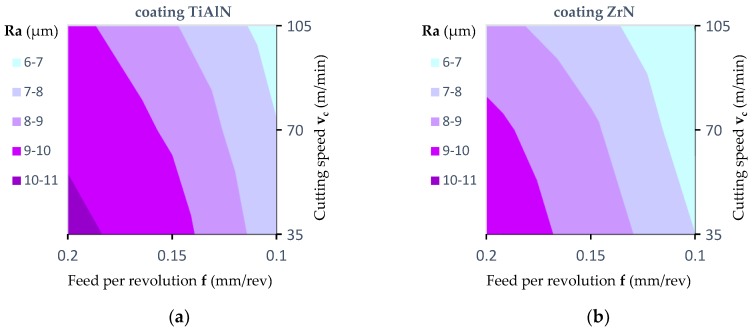
Influence of the feed value on the value of the surface roughness in the inner layer (contour diagrams): (**a**) For a TiAlN coated drill; (**b**) for a ZrN coated drill; (**c**) for an uncoated drill; (**d**) measurement area.

**Figure 18 materials-12-00386-f018:**
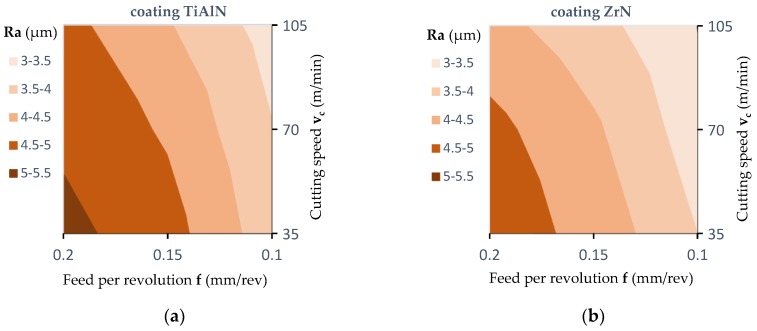
Influence of the feed value on the value of the surface roughness in the outer layer (contour diagrams): (**a**) For a TiAlN coated drill; (**b**) for a ZrN coated drill; (**c**) for an uncoated drill; (**d**) measurement area.

**Figure 19 materials-12-00386-f019:**
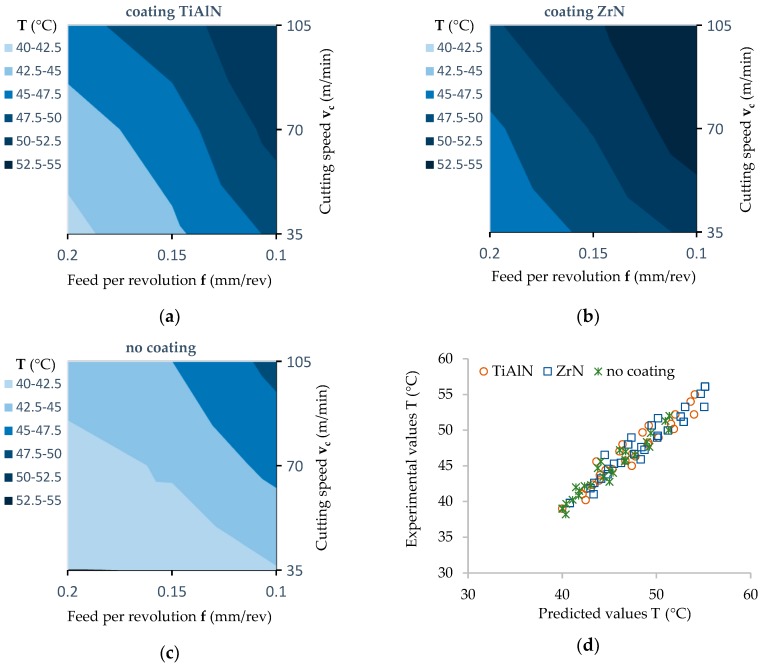
Influence of the feed value on the temperature value (contour diagrams): (**a**) A TiAlN coated drill; (**b**) for a ZrN coated drill; (**c**) for an uncoated drill; (**d**) correlation between the experimental and predicted values of temperature determined for the coated and uncoated tools.

**Table 1 materials-12-00386-t001:** Selected mechanical and physical properties of medium density fiberboard (MDF).

Density (kg/m^3^)	Moisture Content (%)	Bending Strength (MPa)	Elasticity Modulus (MPa)	Thermal Conductivity (W/(m·K))	Thermal Expansion (µm/(m·K))
742	7.2	38	2530	0.3	12

**Table 2 materials-12-00386-t002:** Selected mechanical and physical properties of the coatings.

Type of coating	Coating Temperature (°C)	Hardness (HV)	Thickness (µm)	Coefficient of Friction	Thermal Conductivity (W/(m·K))
ZrN	350–500	2200	2–3	0.4	0.28
TiAlN	400–500	3000	2–3	0.6	2.48

**Table 3 materials-12-00386-t003:** Machining conditions.

Cutting Speed (m/min)	Feed Per Revolution (mm/rev)	Feed Rate (mm/min)	Rotational Speed of Drill (rev/min)
35	0.10	111	1114
35	0.15	167	1114
35	0.20	222	1114
70	0.10	222	2229
70	0.15	334	2229
70	0.20	445	2229
105	0.10	334	3344
105	0.15	501	3344
105	0.20	668	3344

**Table 4 materials-12-00386-t004:** Significance level of the effect of cutting parameters on the average thrust force (F_t_).

Tests Applied	Level of Significance (*p* ≤ 0.05)
cutting speed (v_c_)	0.000
feed per revolution (f)	0.000
coating	0.000
coating × cutting speed	0.997
coating × feed per revolution	0.564

**Table 5 materials-12-00386-t005:** Significance level of the effect of cutting parameters on the average cutting torque (M_c_).

Tests Applied	Level of Significance (*p* ≤ 0.05)
cutting speed (v_c_)	0.000
feed per revolution (f)	0.000
coating	0.000
coating × cutting speed	0.853
coating × feed per revolution	0.734

**Table 6 materials-12-00386-t006:** Setup for the roughness measurement (EN ISO 4288).

Mean Groove Spacing for Periodic Profiles RSm (mm)	Measurement Parameter			
	λ_c_ = l_c_ (mm)	l_n_ (mm)	l_t_ (mm)	r_tip_ (μm)
0.013 < RS_m_ ≤ 0.04	0.08	0.4	0.48	2
0.04 < RS_m_ ≤ 0.13	0.25	1.25	1.5	2
0.13 < RS_m_ ≤ 0.4	0.8	4	4.8	2 or 5
0.4 < RS_m_ ≤ 1.3	2.5	12.5	15	5
1.3 < RS_m_ ≤ 4	8	40	48	10

r_tip_ maximum probe tip radius, l_r_ sampling length, l_n_ evaluation length, l_t_ stylus travel (evaluation length plus start and finish lengths).

**Table 7 materials-12-00386-t007:** Significance level of the effect of cutting parameters on the roughness average (Ra).

Tests Applied	Level of Significance (*p* ≤ 0.05)
cutting speed (v_c_)	0.252
feed per revolution (f)	0.000
coating	0.000
coating × cutting speed	0.999
coating × feed per revolution	0.998

**Table 8 materials-12-00386-t008:** Significance level of the effect of cutting parameters on temperature (T).

Tests Applied	Level of Significance (*p* ≤ 0.05)
cutting speed (v_c_)	0.006
feed per revolution (f)	0.000
coating	0.000
coating × cutting speed	0.998
coating × feed per revolution	0.943

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
