# Peer review of "Experimental Study on Drilling MDF with Tools Coated with TiAlN and ZrN"

_materials, 2019, doi:10.3390/ma12030386_

Round 1
Reviewer 1 Report
1. This paper presents experimental studies on cutting force, surface roughness, and temperature in drilling of medium density fibreboard. The results provide information on the dominant process parameters for the drilling performance. 2. In addition to the cutting force, roughness, and temperature, what is the burr formation at the edge of the drilled holes? Is it a concern in evaluating the process? 3. What is the edge radius of the tool flute? Does it have an effect on the temperature?Author Response
Responses
Reviewer 1.
1. This paper presents experimental studies on cutting force, surface roughness, and temperature in drilling of medium density fibreboard. The results provide information on the dominant process parameters for the drilling performance.
2. In addition to the cutting force, roughness, and temperature, what is the burr formation at the edge of the drilled holes? Is it a concern in evaluating the process?
The burr formation at the edge of the drilled holes is considered in the previous paper of authors (reference no. 34). It is very important topic but we decided to devote a separate article to the effect of tool coating type on the delamination in a drilled hole.
3. What is the edge radius of the tool flute? Does it have an effect on the temperature?
The influence of the edge radius of the tool flute on the temperature of the cutting blade was not analyzed in this paper. Therefore, the drills with a very similar value of this radius were used. The comment and a new figure 11 have been added in the subsection 3.2.
The changes in manuscript have been highlighted in yellow.

Reviewer 2 Report
Study analyses MDF drilling with TiAlN-coated, ZrN-coated and uncoated tools. The study is well presented, with specifics results and good graphics and figures. However, this topic has been studied several times in the bibliography and the results of this work are not clear enough. Therefore I suggest several changes to improve the quality of the paper.
- The information in Figure 7b is not easily readable. I recommend increasing the size of the text or image.
- In figure 6b. The shading of the tool tip makes it difficult to visualize the tool geometry. I suggest removing it or selecting another color.
- Why haven't the typical deviations been represented in figures 12, 13, 16, 17, 18? I think its inclusion may be interesting.
- Figure 12d and 13d cannot be presented without their respective explanations. I recommend that these figures appear later to be mentioned in the text.
- The corresponding tool type is not specified in the footer of figure 12d and 13d.
- Why has Figure 12d represented only the values obtained from the analytical model (1)?
- Contour diagrams could facilitate the understanding of the results. I provide a reference in which these graphs are used.
Bañon, F.; Sambruno, A.; Fernandez-Vidal, S.; Fernandez-Vidal, S.R. One-Shot Drilling Analysis of Stack CFRP/UNS A92024 Bonding by Adhesive. Materials 2019, 12, 160.
- In the conclusions there is no specific mention of the influence of the coating types on the results.
Author Response
Responses
Reviewer 2.
- The information in Figure 7b is not easily readable. I recommend increasing the size of the text or image.
The figure 7b has been enlarged.
- In figure 6b. The shading of the tool tip makes it difficult to visualize the tool geometry. I suggest removing it or selecting another color.
The shading of the tool tip is removed.
- Why haven't the typical deviations been represented in figures 12, 13, 16, 17, 18? I think its inclusion may be interesting.
The figures mentioned are replaced by contour diagrams according to your comment no. 7.
- Figure 12d and 13d cannot be presented without their respective explanations. I recommend that these figures appear later to be mentioned in the text.
The locations of text that mentions the figures have been replaced.
- The corresponding tool type is not specified in the footer of figure 12d and 13d.
The captions have been corrected.
- Why has Figure 12d represented only the values obtained from the analytical model (1)?
The figures 13d, 14d and 19d represented the correlation between experimental and predicted values of specific parameter determined for coated and uncoated tools. The suitable comments have been added in the captions of figures 13d, 14d and 19d.
- Contour diagrams could facilitate the understanding of the results. I provide a reference in which these graphs are used.
Banon, F.; Sambruno, A.; Fernandez-Vidal, S.; Fernandez-Vidal, S.R. One-Shot Drilling Analysis of Stack CFRP/UNS A92024 Bonding by Adhesive. Materials 2019, 12, 160.
The contour diagrams have been added.
- In the conclusions there is no specific mention of the influence of the coating types on the results.
The list of conclusions has been corrected according to your comment.
The changes in manuscript have been highlighted in yellow.

Round 2
Reviewer 2 Report
Dear Authors.
The suggested modifications have been satisfactorily made.
King regards.